# FRICTIONAL Q-LEARNING CONFERENCE SUBMISSIONS

## ABSTRACT

We draw an analogy between static friction in classical mechanics and extrapolation error in off-policy RL, and use it to formulate a constraint that prevents the policy from drifting toward unsupported actions. In this study, we present Frictional Q-learning, a deep reinforcement learning algorithm for continuous control, which extends batch-constrained reinforcement learning. Our algorithm constrains the agent's action space to encourage behavior similar to that in the replay buffer, while maintaining a distance from the manifold of the orthonormal action space. The constraint preserves the simplicity of batch-constrained, and provides an intuitive physical interpretation of extrapolation error. Empirically, we further demonstrate that our algorithm is robustly trained and achieves competitive performance across standard continuous control benchmarks.

## 1 INTRODUCTION

Recently, significant progress has been made in addressing extrapolation error using modern off-policy reinforcement learning (RL) algorithms, as on-policy methods suffer from inherent limitations in data efficiency. In off-policy learning, the agent trains from a replay buffer that aggregates past interactions with the environment. However, this setting introduces a fundamental challenge: extrapolation error arising from distributional shift. When the policy queries state–action pairs that are absent or underrepresented in the dataset, value estimates become unreliable, leading to inaccurate policy updates and cascading errors.

At the core of this challenge lies the difficulty of reliably estimating the value function for policies that select actions outside the buffer's support. Closing this gap has therefore become a central objective. However, most off-policy research has improved algorithms primarily from the perspective of exploration rather than distributional mismatch. While extensive exploration helps the agent cover a broader range of state-action pairs, it can also result in redundant or irrelevant visits to the buffer. To mitigate this issue, prior works such as Batch-Constrained Q-learning (BCQ) (Fujimoto et al., 2019) restrict learned policies to remain close to the data distribution without exploration, thereby reducing the likelihood of selecting out-of-distribution (OOD) actions. While effective, there has been little effort to formally or intuitively explain why such constraints improve stability.

To provide such intuition, we propose a new perspective: interpreting extrapolation error through the lens of friction in physical systems. Static friction resists motion on a slope, opposing the tendency of the body to move toward the horizontal surface, with its strength increasing as the slope angle grows. Analogously, in off-policy RL, unsupported state–action pairs act like high-friction regions in the state–action space. This resistance prevents the policy from converging to the true Markov Decision Process (MDP), which represents the most stable environment. The further the policy deviates from the data distribution, the greater the resistance (extrapolation error) it encounters.

In the same principle, the shift of visitation distribution between the true MDP and the replay buffer can be viewed as the angle of a slope. Just as physical friction slows motion and prevents uncontrolled drift to equilibrium, expanded policy constraints based on imitation of frictional force can prevent the policy's movement toward unsupported regions of the state–action space. In addition to BCQ, which reduces extrapolation error by pulling the policy toward the replay buffer's visitation distribution, we introduce a complementary constraint: pushing the policy away from a heterogeneous visitation distribution constructed with orthogonal actions. This dual objective yields a more robust and stable policy.

Building on these principles, we introduce Frictional Q-learning, which trains a deterministic actor-critic architecture (Konda & Tsitsiklis, 1999) guided by a visitation distribution learned through a contrastive variational autoencoder (cVAE) (Abid & Zou, 2019). FQL optimizes the value function while enforcing dual constraints: proximity to buffer-supported actions ($a$) and distance from orthogonal heterogeneous actions ($v$). To model the local action manifold of the buffer, FQL employs a state-conditioned cVAE, which generates candidate actions aligned with buffer data. The cVAE incorporates augmentable orthogonal actions to accelerate convergence toward the true environments.

Our algorithm not only provides an intuitive interpretation of physics-inspired extrapolation error, but also efficiently finds the state-action space of the optimal policy through convergence in the buffer distribution space and divergence in the orthogonal space without exploration. As a result, our algorithm achieves state-of-the-art performance across multiple continuous-control benchmarks and exceeds the performance of other algorithms, even with high-dimensional work. This suggests that FQL could learn a competitive policy with the distribution of the replay buffer.

## 2 RELATED WORK

Deep Deterministic Policy Gradient (DDPG) (Lillicrap et al., 2015) introduced a deterministic actor-critic framework with policy-gradient updates and soft-target networks. However, policies trained under DDPG often suffered from instability due to overestimation bias. Twin Delayed DDPG (TD3) (Fujimoto et al., 2018) addressed this issue by incorporating twin critic networks and delaying target network updates, effectively reducing bias. While TD3 successfully learned reliable policies in continuous state–action tasks, exploration remained limited under its deterministic actor.

To overcome this limitation, subsequent work shifted toward stochastic policies that encourage broader exploration. Soft Actor–Critic (SAC) (Haarnoja et al., 2018) improved policy diversity and robustness by embedding entropy into soft-policy iteration. Maximum Entropy Reinforcement Learning via Energy-Based Normalizing Flow (MEow) (Chao et al., 2024) further combined actor–critic learning with a flow-based policy, enabling multi-modal action distributions and simplified optimization. Despite these advances, stochastic actor methods are structurally more complex, computationally expensive, and often highly sensitive to hyperparameter tuning.

In parallel, physics-inspired principles have shaped many directions in RL research. Energy-based formulations grounded in Boltzmann distributions have been used to define probabilistic policies (Haarnoja et al., 2017). Particle interactions and repulsive forces promote policy diversity, as in Stein Variational Policy Gradient (SVPG) (Liu et al., 2017). Diffusion processes and their reverses enable trajectory generation and planning (Janner et al., 2022), while continuous-time dynamics have been modeled using stochastic differential equations (SDE), Hamilton–Jacobi–Bellman (HJB) equations, and partial differential equations (PDE) (Wang & Zhou, 2020; Jia & Zhou, 2022; 2023; Kim et al., 2021). Hamiltonian and symplectic dynamics have further been leveraged for stable learning in multi-agent and game-theoretic settings (Balduzzi et al., 2018; Loizou et al., 2020).

Building on batch RL, BCQ provides a principled offline RL approach that mitigates distributional shift from unsupported actions in the dataset and ensures convergence to an optimal policy. Prior studies have highlighted the importance of buffer diversity: for instance, de Bruin et al. (2016) demonstrated that state diversity significantly influences performance, and Isele & Cosgun (2018) showed that agent performance closely tracks the distribution of the replay buffer relative to the test distribution. These findings support the view that extrapolation error originates from buffer distributional shift and plays a central role in off-policy RL performance.

Concretely, BCQ constrains the policy to remain close to actions observed in the dataset. This is achieved by training a generative model, typically a Variational Autoencoder (VAE) (Kingma & Welling, 2013), to approximate the behavior policy and generate candidate actions similar to those in the buffer. A perturbation model then slightly adjusts these generated actions within a bounded range, allowing for limited policy improvement while avoiding large deviations from the data distribution. The perturbed candidate actions are evaluated by the Q-function, and the highest-valued action is selected. While BCQ improves stability and mitigates catastrophic policy failures in offline settings, its performance heavily depends on the quality of the generative model, requires careful tuning of the perturbation module, and remains inherently constrained by the buffer without the ability to explore beyond it.

# 3 BACKGROUND

We consider an agent interacting with a continuous environment in discrete time steps, modeled as an MDP $(\mathcal{S}, \mathcal{A}, p_M(s'|s,a), r, \gamma)$, where $\mathcal{S}$ denotes the state space, $\mathcal{A}$ the action space, $p_M(s'|s,a)$ the transition dynamics, $r(s,a,s')$ the reward function, and $\gamma \in [0,1)$ the discount factor. At each time step $t$, the agent observes a state $s_t \in \mathcal{S}$, selects an action $a_t \in \mathcal{A}$, receives a scalar reward $r_t = r(s_t, a_t, s_{t+1})$, and transitions to the next state $s_{t+1} \sim p_M(\cdot \mid s_t, a_t)$. The objective is to learn a policy $\pi : \mathcal{S} \to P(\mathcal{A})$ that maximizes the expected discounted return $R_t = \sum_{i=t}^{\infty} \gamma^{i-t} r(s_i, a_i, s_{i+1})$, which balances immediate and long-term rewards via $\gamma$.

A policy $\pi$ induces a state visitation distribution $\mu^\pi(s)$ over $\mathcal{S}$. For a given policy, the corresponding action–value function $Q^\pi(s,a)$ is the unique fixed point of the Bellman operator $\mathcal{T}^\pi$, which is a $\gamma$-contraction. Replacing $\mathcal{T}^\pi$ with the optimality operator $\mathcal{T} := \max_a \mathcal{T}^\pi$ yields the optimal value function $Q^\star(s,a) = \max_\pi Q^\pi(s,a)$, and the greedy selection $a^\star = \arg\max_a Q^\star(s,a)$ recovers an optimal policy (Bertsekas, 2008):

$$\mathcal{T}Q(s,a) := \max_a \mathcal{T}^\pi Q(s,a) = \mathbb{E}_{s' \sim p(\cdot|s,a)}\Big[ r(s,a,s') + \gamma \max_{a'} Q(s',a') \Big] \tag{1}$$

Q-learning is used as an off-policy algorithm (Precup et al., 2001); it bootstraps its value estimation with targets that do not depend on the behavior that produced the data. In batch RL, experienced data, $(s,a,r,s') \in \mathcal{B}$, is stored and sampled in a replay buffer $\mathcal{B}$. For high-dimensional continuous state–action spaces, the agent adopts a deterministic actor–critic architecture (Konda & Tsitsiklis, 1999). The critic network $Q_\varphi(s,a)$ approximates the value and is trained on mini-batches with a TD loss and policy gradient (Silver et al., 2014). In contrast, the actor network $\pi_\omega(s)$ is updated to maximize the value, effectively learning actions that approximate $\arg\max_a Q_\omega(s,a)$:

$$\mathcal{L}(\varphi) = \tfrac{1}{2}\, \mathbb{E}_{(s,a,r,s') \in \mathcal{B}}\Big[ \big(r + \gamma \max_{a'} Q_{\varphi^-}(s',a') - Q_\varphi(s,a)\big)^2 \Big], \quad \varphi \leftarrow \varphi - \alpha\, \nabla_\varphi \mathcal{L}(\varphi) \tag{2}$$

$$\nabla_\omega J(\omega) = \mathbb{E}_{s \in \mathcal{B}}\Big[ \nabla_\omega \pi_\omega(s)\, \nabla_{\pi_\omega(s)} Q_\varphi(s, \pi_\omega(s)) \Big], \quad \omega \leftarrow \omega + \beta\, \nabla_\omega J(\omega) \tag{3}$$

To regulate overestimation, both target networks $Q_{\varphi^-}$, $Q_{\omega^-}$ are operated with the $\xi$-weighted ($0 < \xi \ll 1$) average of current parameters from each main and target network after several time steps.

## 3.1 EXTRAPOLATION ERROR

Formally, extrapolation error arises from the distributional shift between the dataset distribution $\mu_\mathcal{B}(s)$ and the state visitation distribution of the current policy $\mu^\pi(s)$. In off-policy RL, transitions $(s,a,r,s')$ are stored in a replay buffer $\mathcal{B}$, and Bellman backups rely on this fixed dataset. However, the target policy may query state–action pairs $(s',a')$ that are rare or absent from $\mathcal{B}$. When the neighborhood of $(s', \pi(s'))$ is sparsely represented in the buffer, the backup target depends on unsupported estimates, and errors can accumulate over successive updates. This compounding effect leads to systematically biased and inaccurate $Q$-value estimates. Moreover, the Bellman operator $\mathcal{T}^\pi$ is defined with respect to the empirical transition dynamics $p_\mathcal{B}(s'|s,a)$ induced by $\mu_\mathcal{B}(s)$, which may be biased relative to the true MDP dynamics $p_M(s'|s,a)$:

$$\mathcal{T}^\pi Q(s,a) \approx \mathbb{E}_{s' \sim \mathcal{B}}\big[ r + \gamma Q(s', \pi(s')) \big] \neq \mathbb{E}_{s' \sim M}\big[ r + \gamma Q(s', \pi(s')) \big]. \tag{4}$$

Standard deep Q-learning updates sample transitions uniformly from $\mathcal{B}$, weighting the loss according to the empirical frequency of $(s,a)$ pairs. If $\mu_\mathcal{B}(s) \neq \mu^\pi(s)$, the learned $Q_\varphi$ provides poor estimates for actions favored by the current policy but rarely represented in $\mathcal{B}$. Reweighting the loss by the likelihood of the current policy does not resolve the problem if high-probability $(s,a)$ pairs under $\pi$ are simply missing from the dataset:

$$\frac{1}{|\mathcal{B}|} \sum_{(s,a,r,s') \in \mathcal{B}} \big\| r + \gamma Q_{\varphi^-}\big(s', \pi(s')\big) - Q_\varphi(s,a) \big\|^2. \tag{5}$$

As a result, only a restricted subset of policies—those close to the behavior policy—can be evaluated reliably in the pure off-policy. When the learned policy selects actions far outside the support of $\mathcal{B}$, training the value function from off-policy data alone can lead to substantial extrapolation error, undermining both policy evaluation and improvement.

## 3.2 STATIC FRICTION

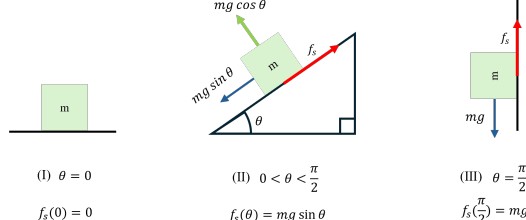

Figure 1: Body of mass $m$ on an inclined plane at angle $\theta$, illustrating force components and static friction.

Consider a body of mass $m$ resting on a plane inclined at an angle $\theta$ relative to the horizontal. The gravitational force $mg$ acts vertically downward and can be decomposed into two components with respect to the plane: a tangential component $mg \sin \theta$ directed downslope, and a normal component $mg \cos \theta$ perpendicular to the surface (Newton, 1833).

When (i) $\theta = 0$ (horizontal surface), the tangential component vanishes since $mg \sin 0 = 0$, meaning no downslope force acts to move the body. In this case, no static friction is required for equilibrium. For (ii) $0 < \theta < \frac{\pi}{2}$, the tangential component $mg \sin \theta$ induces a tendency for the body to slide downslope. Static friction $f_s$ counteracts this motion by acting upslope. The body remains at rest provided that where $\mu_s$ is the coefficient of friction and $N = mg \cos \theta$ is the normal force:

$$mg \sin \theta \;\le\; f_{s,\max} \;=\; \mu_s N \;=\; \mu_s \, mg \cos \theta, \tag{6}$$

In equilibrium, $f_s$ exactly balances $mg \sin \theta$. If the inequality is saturated, static friction reaches its maximum $f_{s,\max}$, beyond which the body begins to slide and kinetic friction takes over. In the extreme case (iii) $\theta = \pi/2$ (vertical surface), the normal force vanishes ($N = 0$). To keep the body stationary, friction would need to counter the full weight, requiring $f_s = mg$ (Coulomb, 1821).

## 4 ALGORITHM

Off-policy deep RL algorithms often select actions without explicitly accounting for the reliability of their value estimates. This can lead to extrapolation error, where OOD actions are incorrectly assigned high values. By contrast, policy evaluation is more reliable within the data-supported regions of the state–action space. To address this, BCQ constrains the learned policy to remain close to actions present in the replay buffer $\mathcal{B}$, thereby mitigating extrapolation error by pulling selected actions toward the buffer-supported distribution (Fujimoto et al., 2019).

We extend this idea by drawing an analogy between extrapolation error and static friction. Specifically, we treat the extrapolation error $\mathcal{E}$ as analogous to the resisting force $f_s$ in mechanics. To capture this effect, we expand the batch-constrained framework by introducing a heterogeneous buffer $\mathcal{H}$, consisting of states $s$ paired with orthonormal actions $v$ ($a \perp v$) relative to the buffer actions $a \in \mathcal{B}$. We define $\theta$ and its complement $\rho = \pi/2 - \theta$ as indicators of the distributional shift between $\mathcal{B}$, $\mathcal{H}$, and the true MDP. Under this formulation, the gravitational force $mg$ corresponds to the extrapolation error $\mathcal{E}_{\pi/2}(s, a)$. Minimizing this error reduces to converging $\theta$ toward zero, effectively aligning the policy with the true MDP:

$$\theta = \arctan\left(\frac{\mathcal{E}_{\mathcal{B}}(s,a)}{\mathcal{E}_{\mathcal{H}}(s,a)}\right) = \arctan\left(\frac{\mathcal{E}_{\theta}(s,a)}{\mathcal{E}_{\rho}(s,a)}\right), \quad 0 \le \theta \le \frac{\pi}{2}. \tag{7}$$

We train a couple of Q-networks, and receive the minimum of their estimations. This update encourages the agent to remain within familiar regions of the state–action space while explicitly reducing extrapolation error. The theoretical underpinnings of FQL rely on assumptions about $\mathcal{H}$ in finite MDPs, enabling an explicit quantification of extrapolation error in this extended framework.

## 4.1 BATCH CONSTRAINED Q-LEARNING

BCQ defines $\mathcal{M}_0$ as the true MDP with transition dynamics $p_0(s'|s,a)$ and initial value estimates $Q_0(s,a)$. Given dataset $\mathcal{B}$, we construct a new MDP $\mathcal{M}_\theta$ that shares the same state and action space as $\mathcal{M}_0$, but whose transitions are induced entirely by $\mathcal{B}$ and augmented with an initial state $s_{\text{init}}$.

Specifically, the transition probabilities are $p_\theta(s'|s,a) = \frac{N(s,a,s')}{\sum_{\tilde{s}} N(s,a,\tilde{s})}$ where $N(s,a,s')$ counts the number of times $(s,a,s')$ appears in $\mathcal{B}$. If no transitions for $(s,a)$ are observed (i.e., $\sum_{\tilde{s}} N(s,a,\tilde{s}) = 0$), the agent transitions to the terminal state $s_{\text{init}}$ with probability 1 and a reward $r(s,a,s_{\text{init}})$ equal to the initialized value $Q_0(s,a)$ is assigned.

**Theorem 1.** Q-learning performed by sampling exclusively from $\mathcal{B}$ converges to the optimal value function under $\mathcal{M}_\theta$.

The tabular extrapolation error in BCQ can then be expressed as the difference between the value function $Q_\theta^\pi$ under $\mathcal{M}_\theta$ and $Q_0^\pi$ under the true MDP $\mathcal{M}_0$:

$$\mathcal{E}_\theta(s,a) = \sum_{s'} \big(p_0(s'|s,a) - p_\theta(s'|s,a)\big)\Big(r(s,a,s') + \gamma \sum_{a'} \pi(a'|s')Q_\theta^\pi(s',a')\Big)$$
$$+ \sum_{s'} p_0(s'|s,a)\gamma \sum_{a'} \pi(a'|s')\mathcal{E}_\theta(s',a') \qquad (8)$$

**Lemma 1.** For all reward functions, $\mathcal{E}_\theta^\pi = 0$ if and only if $p_\theta(s'|s,a) = p_0(s'|s,a)$ for all $s'$ and all $(s,a)$ with $\mu^\pi(s) > 0$ and $\pi(a|s) > 0$.

Lemma 1 implies that a policy derived from $\mathcal{M}_\theta$ must be supported in the same regions of transition probabilities as the true MDP to guarantee convergence. Recovering the true dynamics requires infinitely many samples in stochastic MDPs, whereas in deterministic MDPs, a single transition suffices.

**Theorem 2.** For a deterministic MDP and all reward functions, $\mathcal{E}_\theta^\pi = 0$ if and only if the policy $\pi$ is batch-constrained. Moreover, if $\mathcal{B}$ is coherent, such a policy must exist whenever $s_0 \in \mathcal{B}$.

For BCQ, we require the replay buffer $\mathcal{B}$ to be coherent. We define a buffer $\mathcal{B}$ as coherent if its set of transitions can be arranged into valid trajectories under the environment's dynamics, without requiring states or actions outside of $\mathcal{B}$. This condition is satisfied, for example, when the data are collected as complete trajectories, or when $\mathcal{B}$ contains every state that may be visited.

**Theorem 3.** Under the Robbins–Monro stochastic approximation conditions on the learning rate $\alpha$ and standard sampling requirements, BCQ converges to the optimal value function $Q^*$.

If $\mathcal{B}$ is coherent in a deterministic MDP, BCQ converges to the optimal batch-constrained policy $\pi_\mathcal{B}^*$, which lies strictly within the support of $\mathcal{B}$ and satisfies $Q^{\pi_\mathcal{B}^*}(s,a) \geq Q^\pi(s,a), \forall \pi \in \Pi_\mathcal{B}, \ s,a \in \mathcal{B}$.

**Theorem 4.** For a deterministic MDP and coherent buffer $\mathcal{B}$, along with the Robbins-Monro stochastic convergence conditions on the learning rate $\beta$ and standard sampling requirements on $\mathcal{B}$, BCQ converges to $Q_\mathcal{B}^\pi(s,a)$, where $\pi^*(s) = \arg\max_{a \text{ s.t. } (s,a) \in \mathcal{B}} Q^{\pi_\mathcal{B}}(s,a)$

BCQ ensures convergence by constraining the policy to buffer-supported actions, thereby outperforming any behavioral policy whenever $s_0 \in \mathcal{B}$ (starting from any state within the batch). This guarantee holds without additional assumptions on state–action visitation, aside from coherence.

$$Q(s,a) \leftarrow (1-\alpha)Q(s,a) + \alpha\left(r + \gamma \max_{a' \text{ s.t. } (s',a') \in \mathcal{B}} Q(s',a')\right) \qquad (9)$$

Although BCQ sets $\mathcal{B}$ as a fixed replay buffer, its theoretical guarantees naturally extend to off-policy RL settings, where the buffer is updated concurrently through environment interaction. However, the BCQ introduces a perturbation model to increase the diversity of distribution for the actor without a prohibitive number of samples from the generative model. Empirically, we did not reflect the perturbation model because we consider that it could be replaced with a sample of an improved replay buffer, and it could hinder the training of the actor on off-policy RL.

## 4.2 FRICTIONAL Q-LEARNING

While a batch-constrained policy ensures the convergence of $\mathcal{E}_\theta(s, a)$, our algorithm achieves more robust convergence by additionally considering $\mathcal{E}_\rho(s, a)$. To connect these transition probabilities to the known framework, we treat $\theta$ as the visitation distribution of the orthogonal pair $(a, v)$, assuming reflection symmetry such that $p_\theta(s'|s, a) = p_\rho(s'|s, v)$, where $a \perp v$.

Here, heterogeneity is induced by the degree to which the MDP experiences $v$. The true transition $p_0(s'|s, a)$ corresponds to the optimal MDP $\mathcal{M}_0$, while the most heterogeneous case $\mathcal{M}_{\pi/2}$ corresponds to transitions under $p_{\pi/2}(s'|s, v)$.

**Theorem 5.** For all reward functions, $\mathcal{E}_\rho^\pi$ converges to a biased non-zero, fixed point if $p_\theta(s'|s, v) \neq p_0(s'|s, a), \quad \forall s' \in \mathcal{S}$ with $\mu^\pi(s) > 0$, $\pi(a|s) > 0$, and $\pi(v|s) > 0$.

Since transition probability is a function of $\theta$ and symmetric in $(a, v)$, we can express the extrapolation error as:

$$\mathcal{E}_\rho(s, a) = \sum_{s'} \left( p_0(s'|s, a) - p_\theta(s'|s, v) \right) \left( r(s, a, s') + \gamma \sum_{a'} \pi(a'|s') Q_\rho^\pi(s', a') \right)$$

$$+ p_0(s'|s, a) \gamma \sum_{a'} \pi(a'|s') \mathcal{E}_\rho(s', a') \qquad (10)$$

Theorem 5 implies that if $\pi$ is trained not only to be batch-constrained but also to maintain a prescribed distance from the transition dynamics of an MDP defined by $(s, v)$, then $\pi$ can converge to the true MDP $\mathcal{M}_0$.

**Lemma 2.** Let $\mathcal{A} \subset \mathbb{R}^n$ be a continuous action space with $n \geq 2$ and $\text{rank}(\mathcal{A}) = k$. For any $a \in \mathcal{A}$, there exist orthonormal vectors $v_1, \ldots, v_{k-1} \in \mathcal{A}$ such that $(a, v_i) \neq 0$ for all $i \in \{1, \ldots, k-1\}$.

A trivial orthonormal vector $v$ can always be obtained via a basis shift within the space. Furthermore, $v$ can couple $k-1$ distinct state–action pairs $(s, v_0), \ldots, (s, v_{k-1})$, which enables data augmentation in a cVAE framework.

### 4.2.1 CONTRASTIVE VARIATIONAL AUTOENCODER

We define the target dataset as $\mathcal{D}_t = \{x_i = (s_i, a_i)\}_{i=1}^n$ and construct a background dataset $\mathcal{D}_b = \{b_{i,j} = (s_i, v_{i,j})\}_{\substack{i=1,\ldots,n \\ j=1,\ldots,k-1}}$, $|\mathcal{D}_b| = n(k-1)$, where each $v_{i,j}$ is orthonormal to $a_i$.

Each observation is generated from two independent latent variables via a nonlinear decoder $f$: a *salient* variable $\bar{s} \sim \mathcal{N}(0, I)$ capturing the structure of interest, and an *irrelevant* variable $z \sim \mathcal{N}(0, I)$ capturing nuisance variation. With Gaussian encoders $q_\phi^{\bar{s}}$ and $q_\phi^z$, the evidence lower bounds for a target sample $x_i$ and a background sample $b_{i,j}$ are:

$$\mathcal{L}_t(x_i) = \mathbb{E}_{q_\phi^{\bar{s}} q_\phi^z} \left[ \log f_\theta(a_i|s_i, \bar{s}, z) \right] - \beta \left( \text{KL}\left[ q_\phi^{\bar{s}}(\bar{s}|x_i) \,\|\, p(\bar{s}) \right] - \text{KL}\left[ q_\phi^z(z|x_i) \,\|\, p(z) \right] \right) \quad (11)$$

$$\mathcal{L}_b(b_{i,j}) = \mathbb{E}_{q_\phi^z} \left[ \log f_\theta(v_{i,j}|s_i, 0, z) \right] - \beta \, \text{KL}\left[ q_\phi^z(z|b_{i,j}) \,\|\, p(z) \right]. \quad (12)$$

We further incorporate a total-correlation term $\text{TC}(\bar{s}, z)$:

$$\text{TC}(\bar{s}, z) = \log \frac{D_\psi(\bar{s}, z)}{1 - D_\psi(\bar{s}, z)},$$

where the discriminator $D_\psi$ promotes statistical independence between $\bar{s}$ and $z$. The overall training objective, with hyperparameter $\beta > 0$ balancing KL divergence against reconstruction fidelity, is:

$$\max_\phi \frac{1}{n} \sum_{i=1}^n \left[ \mathcal{L}_t(x_i) - \text{TC}\left( q_\phi^{\bar{s}}(x_i), q_\phi^z(x_i) \right) \right] + \frac{1}{n(k-1)} \sum_{i=1}^n \sum_{j=1}^{k-1} \mathcal{L}_b(b_{i,j}). \quad (13)$$

All networks are trained jointly using mini-batch stochastic gradient descent. Biases are removed, and ReLU activations are employed so that an input of 0 remains unchanged throughout the computation. The contrastive architecture separates salient and irrelevant factors, then cVAE projects the transition dynamics of $\mathcal{M}_\theta$ into a latent space by contrasting $a \in \mathcal{B}$ with its orthogonal counterpart $v$. Moreover, orthonormal bases provide background samples for cVAE, serving as a form of data augmentation to improve generalization.

### 4.2.2 BATCH-CONSTRAINED CONTRASTIVE REINFORCEMENT LEARNING

---

**Algorithm 1** Frictional Q-learning (FQL)

---

Initialize critic networks $Q_{\varphi_1}$, $Q_{\varphi_2}$ with random parameters $\varphi_1, \varphi_2$
Initialize actor network $\mu_\omega$ with random parameters $\omega$
Initialize contrastive VAE $G_\zeta = \{q_\phi^{\bar{s}}, q_\phi^z, f_\theta\}$ with random parameters $\zeta$
Initialize target networks $\varphi_1^- \leftarrow \varphi_1$, $\varphi_2^- \leftarrow \varphi_2$, $\omega^- \leftarrow \omega$
Initialize replay buffer $\mathcal{B}$
**for** $m = 1$ **to** $M$ **do**
  Initialize a random process $\mathcal{N}$ for action exploration
  Receive initial observation state $s_{\text{init}}$
  **for** $t = 1$ **to** T **do**
    Select action $a_t = \mu_\omega(s_t) + \mathcal{N}_t$ according to the current policy and exploration noise
    Execute $a_t$, compute orthonormal action $v_t$, observe reward $r_t$ and next state $s_{t+1}$
    Store $(s_t, a_t, r_t, s_{t+1})$ in $\mathcal{B}$
    Sample minibatch $(s_i, a_i, v_i, r_i, s_{i+1})$ of $N$ transitions from $\mathcal{B}$
    Set $y_{i+1} = \max\limits_{d \in \{1,\dots,c\}} \min\limits_{\ell \in \{1,2\}} Q_{\varphi_\ell^-}\big(s_{i+1}, \tilde{a}_{i+1}^{(d)}\big)$, $\{\tilde{a}_{i+1}^{(d)}\}_{d=1}^k \sim f_\theta(s_{i+1}, \bar{s}, z)$
    Update critics :

    $$\mathcal{L}_Q(\varphi_1, \varphi_2) \;=\; \frac{1}{N}\sum_{i=1}^n \sum_{\ell=1}^2 \Big(Q_{\varphi_\ell}(s_{i+1}, a_{i+1}) - y_i\Big)^2$$

    Update actor:

    $$J_\mu(\omega) \;=\; -\frac{1}{N}\sum_{i=1}^n Q_{\varphi_1}\big(s_i, \tilde{a}_{i+1}\big), \;\; \tilde{a}_{i+1} \sim f_\theta(s_i, \bar{s}, z)$$

    Update cVAE with $v_i = v_{i,j^*}$, $\quad j^* = \arg\min\limits_{j \in \{1,\dots,k\}} Q_{\varphi_1}(s, v_{i,j})$

    $$\mathcal{L}_G(\zeta) = \frac{1}{N}\sum_{i=1}^n \Big[\mathcal{L}_t(s_i, a_i) \;-\; \mathrm{TC}\big(q_\phi^{\bar{s}}(s_i, a_i), q_\phi^z(s_i, a_i)\big) + \mathcal{L}_b\big(s_i, v_i\big)\Big]$$

    Update targets:

    $$\varphi_\ell^- \leftarrow \tau\,\varphi_\ell + (1-\tau)\,\varphi_\ell^-, \quad \ell = 1, 2, \qquad \omega^- \leftarrow \tau\,\omega + (1-\tau)\,\omega^-$$

  **end for**
**end for**

---

FQL can be viewed as an off-policy, batch-constrained, contrastive RL algorithm. The cVAE generates candidate actions $\tilde{a}$ aligned with the actions in the replay buffer $\mathcal{B}$. A critic with double Q-networks (Fujimoto et al., 2018) evaluates these candidates, selecting the maximum-valued action for the current state while also providing cVAE with the most undervalued orthonormal candidates.

To preserve the batch-constrained property, we introduce a state-conditioned marginal density $P_{\mathcal{B}}^G(a|s)$, which quantifies how likely an action $a$ is in state $s$ relative to the state–action pairs in the replay buffer $\mathcal{B}$. A policy defined as $\pi^\star(s) = \arg\max_a P_{\mathcal{B}}^G(a|s)$ avoids extrapolation errors by restricting action choices to those supported by the data. We approximate this density using a cVAE $G_\zeta(s)$, and treat sampled actions $\tilde{a} \sim G_\zeta(s)$ as surrogates for policy estimation:

$$\pi(s) = \arg\max_{i \in \{1,\dots,m\}} Q_\varphi\big(s, \tilde{a}_i\big), \quad \{\tilde{a}_i\}_{i=1}^m \sim f_\theta(s, \bar{s}, z) \tag{14}$$

Here, the latent variables $\bar{s}, z \sim \mathcal{N}(0, I)$ are decoded together with the current state $s$ to produce candidate actions $\tilde{a} = G_\zeta(s, \bar{s}, z)$. The policy is then obtained by sampling multiple candidates and selecting the one with the highest estimated return from the Q-network $Q_\varphi$.

# 5 RESULTS

We evaluated the proposed FQL method on high-dimensional continuous control tasks from Mu-JoCo (Todorov et al., 2012) via the Gymnasium benchmark suite (Towers et al., 2024), a widely adopted robotic simulation platform for reinforcement learning research. Our evaluation focuses on both stability and sample efficiency, with particular focus on extrapolation error. We compare FQL against a diverse set of actor–critic baselines covering both stochastic and deterministic policies. Each algorithm was trained independently with five different random seeds to account for stochasticity in initialization and environment dynamics using an NVIDIA RTX A6000 (48GB). For each environment, we report mean episode returns every 10,000 steps, along with standard deviations across seeds. Baselines were reproduced using Stable-Baselines3 (SB3) (Raffin et al., 2021); for MEow, which is not implemented in SB3, we used the official code from the original authors. All hyperparameters not explicitly mentioned follow the original implementations in SB3's refinement. Training required approximately 5.6 hours per million steps for the critic and 6.1 hours per million steps for the augmentation process, with each GPU running all seed sessions concurrently.

## 5.1 COMPARISON

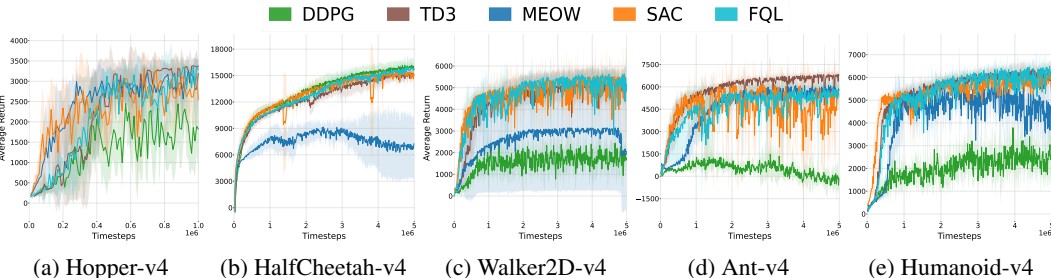

(a) Hopper-v4    (b) HalfCheetah-v4    (c) Walker2D-v4    (d) Ant-v4    (e) Humanoid-v4

Figure 2: Average return (solid line) and standard deviation (shaded area) across five independent runs with different random seeds in continuous-control environments.

Performance was assessed with three complementary metrics: Step, Seed, and Final. **Step** denotes the maximum of the five-seed average evaluated at each timestep, **Seed** is the mean of the maximum values obtained from each seed, and **Final** corresponds to the mean performance at the last evaluation step. While excelling in a single metric does not establish state-of-the-art performance, achieving strong results across all three metrics indicates robustness and superiority. Full numeric results are provided in Appendix 3.

As shown in Figure 2, FQL outperformed baseline methods across multiple tasks with a large margin, achieving state-of-the-art performance on Walker2D-v4 and Humanoid-v4. Notably, Humanoid-v4 is typically considered favorable for stochastic policies but challenging for deterministic ones, highlighting FQL's strength. By contrast, on lower-dimensional tasks such as Hopper-v4, performance decreased, which we attribute to the limited effectiveness of evaluating only a few candidates for the constraint and background datasets. On HalfCheetah-v4, where state–action samples are more reliably collected, FQL underperformed relative to DDPG, although the margin was small and narrower than that between DDPG and the next best algorithm, SAC. Across all environments, FQL demonstrated rapid convergence and stable long-term performance.

Due to the stochasticity of its latent variables, FQL achieves exploration behavior similar to stochastic actors, while retaining the efficiency and stability of a deterministic framework. Furthermore, FQL exhibited markedly narrower standard deviations across metrics compared to baselines in tasks where it reached state-of-the-art results. We attribute such robustness to the inherent mathematical stability of batch-constrained Q-learning.

Finally, our expanded frictional constraints with orthonormal vectors were easily satisfied using a state-conditioned network. The final buffer distributions confirmed that the action $a$ and orthonormal vector $v$ remained fully separated, as shown in Appendix 5. This contrasted sharply with results obtained using a zero vector as the heterogeneous action, validating our assumption that orthonormal vectors provide an appropriate and effective choice for constructing heterogeneous actions.

## 5.2 ABLATION

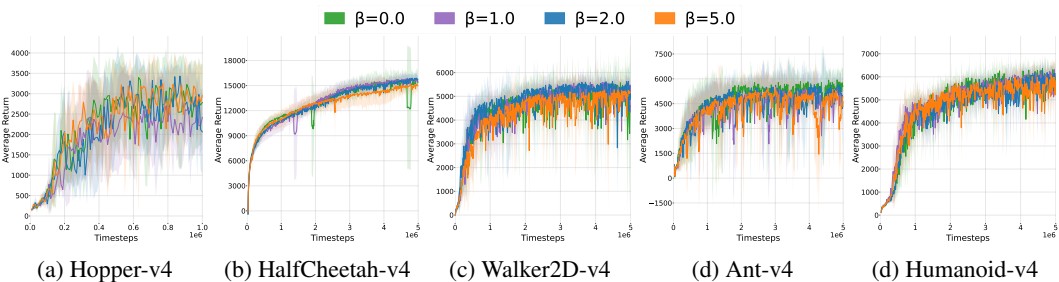

(a) Hopper-v4    (b) HalfCheetah-v4    (c) Walker2D-v4    (d) Ant-v4    (d) Humanoid-v4

Figure 3: Average return (solid line) and standard deviation (shaded area) across five runs with different seeds, showing the sensitivity of FQL performance to the $\beta$ parameter, which balances salient and irrelevant feature distributions.

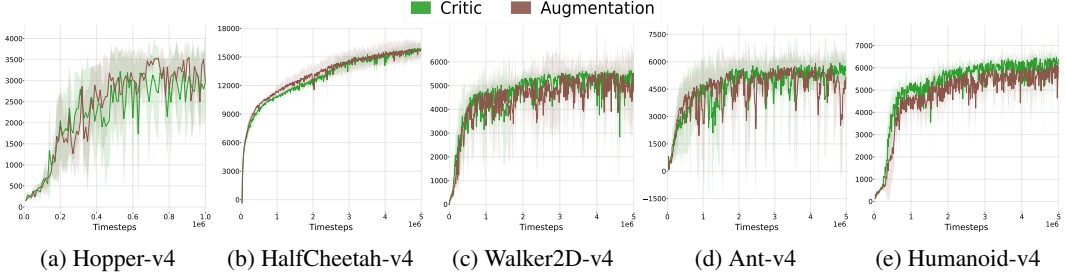

(a) Hopper-v4    (b) HalfCheetah-v4    (c) Walker2D-v4    (d) Ant-v4    (e) Humanoid-v4

Figure 4: Average return (solid line) and standard deviation (shaded area) across five runs with different seeds, comparing Critic-based and Augmentation-based methods for incorporating orthonormal vectors.

Additionally, we conducted an ablation study to evaluate the sensitivity of the cVAE architecture and the effectiveness of orthonormal vectors under different hyperparameter settings and design choices.

**Beta**. Figure 3 shows that no single $\beta$ value consistently yielded the best performance across all tasks. Instead, each environment exhibited a task-specific optimal $\beta$. For instance, in HalfCheetah-v4, only the appropriate $\beta$ value avoided sudden instability or performance degradation, whereas other values led to divergence or poor returns. These results suggest that training does not necessarily require forcing salient and irrelevant features to converge to an isotropic normal distribution, but rather benefits from task-dependent tuning of $\beta$.

**Method**. We compared two strategies for incorporating orthonormal vectors: (i) evaluating them through the Critic and (ii) treating them as background samples with Augmentation. As shown in Figure 4, the Critic-based approach proved generally more effective and stable than Augmentation, with faster convergence and reduced variance across tasks except for Hopper-v4.

## 6 CONCLUSION

We introduce FQL, which applies the concept of static friction to mitigate extrapolation error in batch-constrained Q-learning through a contrastive generative model. FQL provides both an intuitive analogy and an effective framework for continuous control tasks in off-policy RL. Our theoretical analysis shows that the proposed constraints lead to convergence toward the optimal policy, and empirical results demonstrate consistent improvements over strong stochastic baselines. FQL not only enhances performance but also improves robustness and stability, highlighting the potential of physics-inspired formulations to address fundamental challenges in RL. Nonetheless, the stochasticity of the state-conditioned generative distribution can occasionally destabilize training and hinder the critic's ability to evaluate candidate actions reliably. Developing stabilization techniques for this component will be an important direction for future work. All experimental results, proofs, and hyperparameter details are provided in the Appendix.

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

## A PROOFS

### A.1 EQUATION 7

**Definition 1.** A buffer is defined as $(s, a, s') \in \mathcal{B}$ and a heterogeneous buffer as $(s, v, s') \in \mathcal{H}$, unless $s'$ is terminal.

**Definition 2.** The extrapolation error of $\mathcal{B}$, denoted $\mathcal{E}_\theta(s, a)$, is defined as the static frictional force $mg \sin \theta$.

**Definition 3.** The maximum heterogeneous extrapolation error is defined as $\mathcal{E}_{\pi/2}(s, a) = mg$.

**Definition 4.** The extrapolation error of $\mathcal{H}$, denoted $\mathcal{E}_{(\pi/2-\theta)}(s, a)$, is defined as $mg \sin(\pi/2 - \theta)$.

Replacing $mg$ by $\mathcal{E}_{\pi/2}(s, a)$ allows us to eliminate $mg$ and derive the following relations: $\mathcal{E}_\theta(s, a) = \mathcal{E}_{\pi/2}(s, a) \sin \theta$ and $\mathcal{E}_{(\pi/2-\theta)}(s, a) = \mathcal{E}_{\pi/2}(s, a) \sin(\pi/2 - \theta)$. Thus,

$$\frac{\mathcal{E}_\theta(s, a)}{\mathcal{E}_{(\pi/2-\theta)}(s, a)} = \frac{mg \sin \theta}{mg \sin(\pi/2 - \theta)} = \tan \theta$$

Taking the inverse trigonometric function yields $\theta = \arctan\left(\frac{\mathcal{E}_\theta(s,a)}{\mathcal{E}_{(\pi/2-\theta)}(s,a)}\right) \in [0, \pi/2]$

### A.2 THEOREM 5

**Theorem 5.** For all reward functions, $\mathcal{E}_\rho^\pi$ converges to a biased non-zero, fixed point if $p_\theta(s'|s, v) \neq p_0(s'|s, a)$, $\quad \forall s' \in \mathcal{S}$ with $\mu^\pi(s) > 0$, $\pi(a|s) > 0$, and $\pi(v|s) > 0$.

Since the transition probability is a function of $\theta$ and symmetric in $(a, v)$, we can express the extrapolation error transformation as:

$$
\begin{aligned}
\mathcal{E}_\rho(s, a) &= Q_0^\pi(s, a) - Q_\rho^\pi(s, a) \\
&= \sum_{s'} p_0(s'|s, a)\left( r(s, a, s') + \sum_{a'} \pi(a'|s')Q_0^\pi(s', a') \right) \\
&\quad - \sum_{s'} p_\rho(s'|s, a)\left( r(s, a, s') + \sum_{a'} \pi(a'|s')Q_\rho^\pi(s', a') \right) \\
&= \sum_{s'} \big(p_0(s'|s, a) - p_\rho(s'|s, a)\big)\left( r(s, a, s') + \sum_{a'} \pi(a'|s')Q_\rho^\pi(s', a') \right) \\
&\quad + p_0(s'|s, a)\gamma \sum_{a'} \pi(a'|s')\mathcal{E}_\rho(s', a') \\
&= \sum_{s'} \big(p_0(s'|s, a) - p_\rho(s'|s, a)\big)\left( r(s, a, s') + \sum_{a'} \pi(a'|s')Q_\rho^\pi(s', a') \right) \\
&\quad + p_0(s'|s, a)\gamma \sum_{a'} \pi(a'|s')\mathcal{E}_\rho(s', a') \\
&= \sum_{s'} \big(p_0(s'|s, a) - p_\theta(s'|s, v)\big)\left( r(s, a, s') + \gamma \sum_{a'} \pi(a'|s')Q_\rho^\pi(s', a') \right) \\
&\quad + p_0(s'|s, a)\gamma \sum_{a'} \pi(a'|s')\mathcal{E}_\rho(s', a')
\end{aligned}
$$

**Remark 1.** For a discounted MDP with $\gamma \in [0, 1)$ and bounded rewards $|r(s, a, s')| \leq R_{\max}$, consider a fixed policy $\pi$ and define the extrapolation error $\mathcal{E}_\rho(s, a) = Q_0^\pi(s, a) - Q_\rho^\pi(s, a)$ where $Q_0^\pi$ is evaluated with the true kernel $p_0(\cdot|s, a)$ and $Q_\rho^\pi$ with an approximate kernel $p_\rho(\cdot|s, a)$. Then,

$$\|\mathcal{E}_\rho\|_\infty \leq \frac{2 \sup_{s,a} \text{TV}\big(p_0(s'|s, a), p_\rho(s'|s, a)\big)}{(1 - \gamma)^2} R_{\max}$$

Furthermore, the angular constraint satisfies

$$\theta \; \leq \; \arctan \frac{\sup_{s,a} \text{TV}\big(p_0(s'|s,a),\, p_\theta(s'|s,a)\big)}{\sup_{s,a} \text{TV}\big(p_0(s'|s,a),\, p_\rho(s'|s,a)\big)} \; \leq \; \frac{\pi}{4}$$

The theorem introduces an upper bound on the angle that quantifies the discrepancy in terms of total variation ratios. The bound saturates at $\pi/4$ when the transition probability of an action and its orthogonal counterpart exhibit identical deviations from the buffer distribution. Unlike classical bounds, this angle-based constraint is independent of both the discount factor $\gamma$ and the reward scale $R_{\max}$. By extending the constraint to orthogonal actions, the result directly captures model–environment mismatch as a dynamical constraint. This provides a theoretical guarantee that extrapolation error can be stably controlled up to $\pi/4$ purely through buffer design and action-space constraints, without reliance on environment-specific scaling factors.

**Proof.** We start from the standard Bellman error decomposition where $b(s,a) \coloneqq \sum_{s'}\big(p_0 - p_\rho\big)(s'|s,a)\,\Big(r(s,a,s') + \gamma V_\rho^\pi(s')\Big)$. Since $|r(s,a,s')| \leq R_{\max}$ and $|V_\rho^\pi(s)| \leq R_{\max}/(1-\gamma)$, each term satisfies $\big|r(s,a,s') + \gamma V_\rho^\pi(s')\big| \leq \frac{R_{\max}}{1-\gamma}$. Hence $|b(s,a)| \leq \frac{\|p_0(s'|s,a) - p_\rho(s'|s,a)\|_1}{1-\gamma} R_{\max} = \frac{2\,\text{TV}\big(p_0(s'|s,a), p_\rho(s'|s,a)\big)}{1-\gamma} R_{\max}$. Writing the error equation in operator form $\mathcal{E}_\rho = (I - \gamma P_0\Pi)^{-1} b$ where $P_0\Pi$ is the policy-induced Markov operator. Since $\|P_0\Pi\|_\infty = 1$, the Neumann series gives $\big\|(I - \gamma P_0\Pi)^{-1}\big\|_\infty \leq \frac{1}{1-\gamma}$

## A.3 LEMMA 2

**Lemma 2.** Let $\mathcal{A} \subset \mathbb{R}^n$ be a continuous action space with $n \geq 2$ and $\text{rank}(\mathcal{A}) = k$. For any $a \in \mathcal{A}$, there exist orthonormal vectors $v_1, \ldots, v_{k-1} \in \mathcal{A}$ such that $(a, v_i) \neq 0$ for all $i \in \{1, \ldots, k-1\}$.

We define the action space $\mathcal{A}$ over a continuous range $[x, y]$. This space can be shifted to $\mathcal{A}' = [x - S, y - S] = [(x - y)/2, (y - x)/2] = [-r, r]$ where $\mathcal{S} = (x + y)/2$ and $r = (y - x)/2$.

Next, we scale any orthonormal vector $v \in [-1, +1]$ by $r$, obtaining $rv = [-r, +r]$. Hence, after shifting the space and scaling the vectors, we can always guarantee the existence of orthonormal vectors $rv \in A'$.

During FQL training, the algorithm computes the shift $\mathcal{S}$, applies scaling and shifting to normalize the action space, and then recovers the shift when mapping actions back to the environment.

# B EXPERIMENT

## B.1 HYPERPARAMETER

Tables 1 and 2 summarize the common and environment-specific hyperparameters used in our experiments. The latent dimension of the cVAE is set to twice the action dimension.

Table 1: Common hyperparameters used in all experiments

| Parameter | Value |
|---|---|
| Shared optimizer | Adam |
| Actor Learning rate | $3 \times 10^{-4}$ |
| cVAE Learning rate | $1 \times 10^{-3}$ |
| Discount factor | 0.99 |
| Replay buffer size | $10^6$ |
| Number of hidden layers (All networks) | 2 |
| Number of hidden units per layer (Actor & Critic) | 256 |
| Number of hidden units per layer (cVAE) | 512 |
| Network Bias (cVAE) | False |
| Number of samples per minibatch | 256 |
| Nonlinearity | ReLU |
| Target update interval | 2 |
| Gradient steps | 1 |

Table 2: Environment-specific hyperparameters for FQL

| Environment | State Dim | Action Dim | Critic Learning rate | Beta $\beta$ |
|---|---|---|---|---|
| Hopper-v4 | 11 | 3 | 3e-4 | 5.0 |
| HalfCheetah-v4 | 17 | 6 | 1e-3 | 2.0 |
| Walker2D-v4 | 17 | 6 | 1e-3 | 2.0 |
| Ant-v4 | 105 | 8 | 1e-3 | 0.0 |
| Humanoid-v4 | 348 | 17 | 3e-4 | 0.0 |

## B.2 DISTRIBUTION

Figure 5 presents a density histogram of the replay buffer for Humanoid-v4 trained with FQL, comparing original actions (blue) and their orthonormal counterparts (orange). The results reveal clear distributional differences between heterogeneous action pairs. Importantly, each orthonormal action is not aligned with a single coordinate axis but is instead constructed to satisfy orthogonality with respect to the original action.

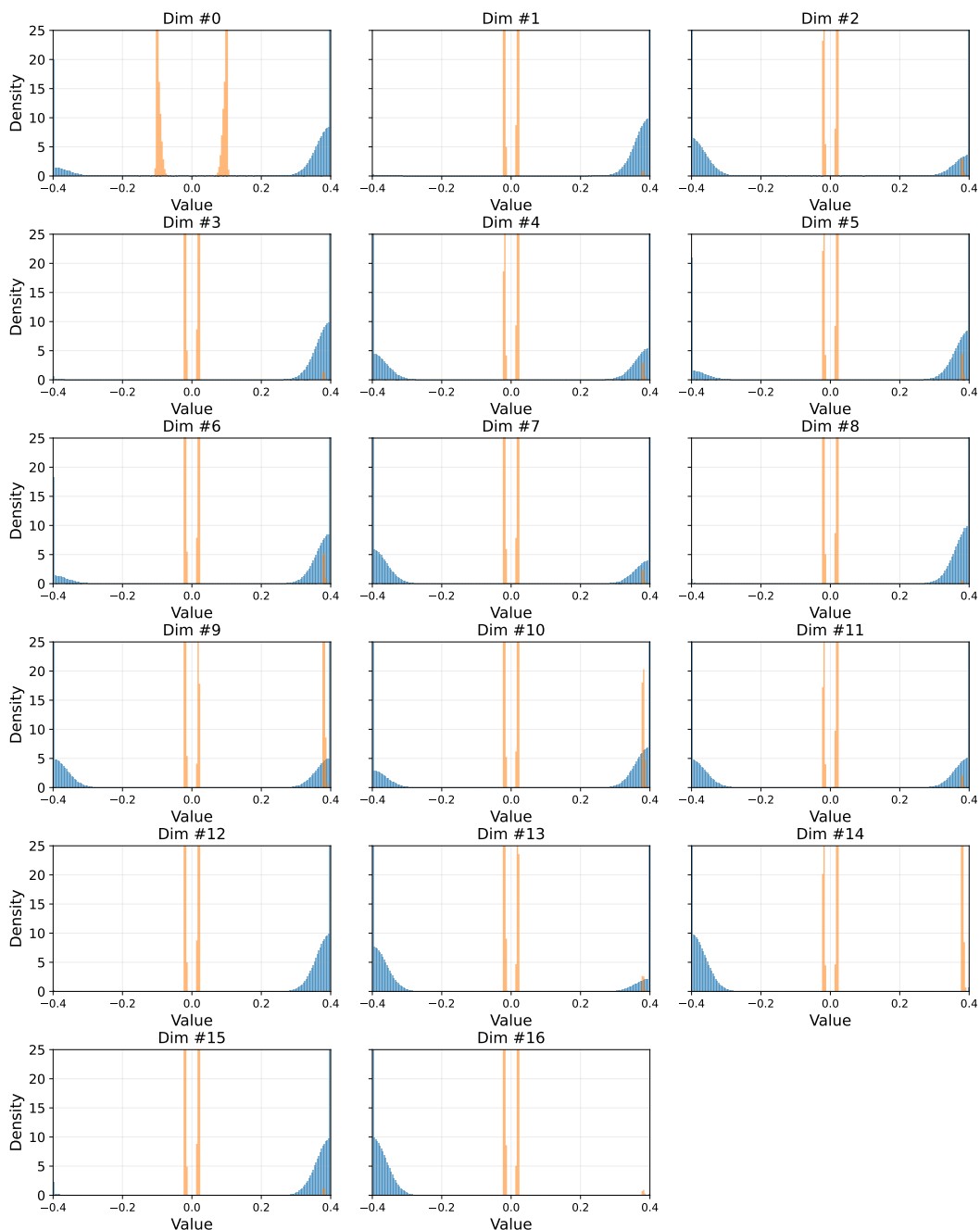

Figure 5: Density distribution of replay buffer actions (blue) and orthonormal actions (orange) in Humanoid-v4.

## B.3 PERFORMANCE

Table 3 compares evaluation metrics across algorithms and environments. The best performing algorithm is marked in **bold** and underlined, and the second-best is underlined.

Table 3: Comparison of evaluation metrics across algorithms and environments

| Algorithm | Metric | Hopper-v4 | HalfCheetah-v4 | Walker2D-v4 | Ant-v4 | Humanoid-v4 |
|---|---|---|---|---|---|---|
| FQL | Step | 3399.07 ± 134.65 | 15884.54 ± 731.41 | **5634.72 ± 135.34** | 5939.56 ± 814.53 | **6486.43 ± 144.90** |
| | Seed | **3834.90 ± 100.50** | 16138.97 ± 591.28 | **5810.95 ± 165.41** | 6220.44 ± 637.83 | **6713.75 ± 103.89** |
| | Final | 2560.17 ± 792.37 | 15764.84 ± 478.06 | **5144.50 ± 1264.19** | 5543.07 ± 1108.04 | **6369.02 ± 193.65** |
| SAC | Step | 3312.25 ± 295.90 | 15476.83 ± 244.64 | 5554.61 ± 452.25 | 6139.32 ± 180.33 | 6302.05 ± 412.21 |
| | Seed | 3834.12 ± 130.31 | 15652.44 ± 118.17 | 5639.36 ± 456.15 | 6413.75 ± 183.03 | 6463.46 ± 322.07 |
| | Final | 2256.28 ± 541.97 | 15100.17 ± 818.84 | 4815.50 ± 1698.39 | 5378.26 ± 1017.06 | 5682.62 ± 327.95 |
| MEOW | Step | 3260.83 ± 234.36 | 9157.70 ± 887.61 | 3147.91 ± 2881.89 | 6116.25 ± 461.82 | 5939.99 ± 729.92 |
| | Seed | 3565.67 ± 131.83 | 9914.46 ± 756.97 | 3499.34 ± 2531.89 | 6361.68 ± 160.68 | 6609.30 ± 693.20 |
| | Final | 2692.54 ± 467.78 | 7241.62 ± 3722.44 | 1916.23 ± 2524.40 | 5752.51 ± 575.51 | 4650.22 ± 1168.05 |
| TD3 | Step | 3718.06 ± 55.87 | 15342.75 ± 627.81 | 5247.39 ± 708.99 | 6832.46 ± 212.72 | 6355.69 ± 488.14 |
| | Seed | 3797.88 ± 58.92 | 15663.80 ± 706.26 | 5335.39 ± 693.30 | 6944.11 ± 172.92 | 6524.20 ± 401.63 |
| | Final | 3140.13 ± 678.48 | 15120.68 ± 517.59 | 5104.41 ± 787.84 | 6800.51 ± 136.22 | 5373.53 ± 1879.89 |
| DDPG | Step | 2447.75 ± 412.87 | 16172.92 ± 1024.99 | 2440.47 ± 676.74 | 1482.48 ± 1110.05 | 3789.23 ± 535.20 |
| | Seed | 3553.73 ± 41.82 | 16363.82 ± 949.05 | 3812.20 ± 538.97 | 2512.60 ± 1037.49 | 4816.93 ± 326.80 |
| | Final | 1567.25 ± 244.41 | 15836.04 ± 508.79 | 1744.66 ± 436.61 | -157.27 ± 691.26 | 2618.13 ± 749.56 |

