# OpenReview forum: "Frictional Q-Learning"
_ICLR.cc/2026/Conference — ICLR 2026 Conference Withdrawn Submission_

### Official Review · Reviewer_777R · 2025-10-28

**Soundness:** 2
**Presentation:** 2
**Contribution:** 2
**Rating:** 2
**Confidence:** 2

**Summary:**

The paper investigates extrapolation error in off-policy reinforcement learning, which arises when function approximators (e.g., neural networks) estimate the value of state–action pairs that are insufficiently represented in the replay buffer, or more generally, when there is a distribution mismatch between the behavior policy and the replay data.

Building on [1], the authors draw an analogy between extrapolation error and static friction, arguing that mitigating this error is analogous to reducing the effective slope angle of a surface.

The proposed approach is evaluated on standard Gym MuJoCo benchmark tasks and compared against commonly used baselines for continuous control.

[1] Off-Policy Deep Reinforcement Learning without Exploration; Scott Fujimoto, David Meger, Doina Precup

**Strengths:**

The idea of linking numerical value estimates in reinforcement learning to physical concepts—friction in this case—is novel and potentially insightful.

**Weaknesses:**

While the idea of drawing an analogy between extrapolation error and friction is conceptually intriguing, the current presentation of this analogy lacks clarity and a strong motivating rationale. It is not yet clear why such a physical interpretation is necessary or beneficial, nor how exactly the components of extrapolation error correspond to the physical forces described.

Additionally, I am not fully convinced by the empirical evaluation.

First, the learning curves in Figure 2 do not clearly demonstrate a significant advantage of FQL over standard baselines. Even in Humanoid, where FQL attains the highest final performance, its confidence intervals overlap substantially with those of TD3 and SAC, which weakens the claim that it consistently outperforms existing methods. I encourage the authors to broaden their evaluation to include more diverse continuous control tasks, such as those from the DeepMind Control Suite, in order to better highlight the strengths of the proposed approach.

Second, given that the primary focus is on mitigating extrapolation error, I would expect experiments in an offline RL setting where no additional data collection is permitted. The current experiments appear to be entirely online, where the agent can continuously gather new samples, thereby shifting the data distribution in a way that naturally reduces extrapolation error. Under such an uncontrolled data-collection regime, it is difficult to isolate and properly assess the contribution of the proposed technique.

**Questions:**

The primary question I would like the authors to clarify concerns the theoretical link between static friction and extrapolation error, which appears to be the central contribution of the paper. Although Section 4 offers a narrative description, the underlying mechanics remain unclear to me.

More specifically, in relation to Section 3.2, what are the exact counterparts of gravitational force ($mg$), static friction ($mg \sin \theta$), and the normal force ($mg \cos \theta$) in terms of extrapolation error? How should we interpret the angle $\theta$ in the context of off-policy value estimation? More fundamentally, what justifies this analogy in the first place? In high-dimensional spaces, there are infinitely many vectors orthogonal to a given vector—for instance, $(1,0,0)$ is orthogonal to all vectors of the form $(0,y,z)$. How are the orthogonal high-dimensional actions constructed in the method, and is there a principled reason behind selecting a specific orthogonal direction? What is the conceptual motivation for doing so?

I would appreciate a deeper and more rigorous explanation of this core idea. With greater clarity and justification, I could evaluate the work more objectively and potentially raise my score. Specifically, if the core idea is sound and insightful, I would be inclined to raise my score even if the authors are unable to provide additional empirical results during the rebuttal period.

---

### Official Review · Reviewer_nFCU · 2025-10-31

**Soundness:** 1
**Presentation:** 1
**Contribution:** 1
**Rating:** 0
**Confidence:** 4

**Summary:**

This paper presents an instantiation of batch-constrained Q learning, where the Q values select actions that are generated from a CVAE that learns to output buffer actions over their synthetic orthogonal counterparts. Like BCQ, constraining updates like this prevents incorporating spurious values in the backup. As I understand, the key algorithmic difference is in how the generative action model is trained, most importantly the contrastive term, but also a total-correlation term. This algorithm performs competitively, though not exceptionally, compared with continuous control baselines.

**Strengths:**

Contrastive learning is a reasonable way to improve the generative modeling component of BCQ.

**Weaknesses:**

I had an extremely hard time understanding this paper, and I feel some parts of this paper were constructed in bad faith.

**To start off, Section 4.1 is almost word-for-word taken from 4.1 of the original BCQ paper (https://arxiv.org/pdf/1812.02900 top of page 5). It’s only mildly, and confusingly, reworded. Without very clear attribution of results, more than beginning the section with "BCQ defines", this is not acceptable.**

In addition,

* Very central terms are undefined. For example “extrapolation error” is only defined recursively (and confusingly) in Equation 8, and maybe in the appendix.
* Theorem 5 is very confusing, I don’t really see a proof of the claim, just a long equality? I’m struggling to understand the relationship between Theorem 5 / Remark 1, which are in terms of transition dynamics, with the work above, which is in terms of action distributions.
* The analogy to friction, to me, was just confusing. What is the relation to “static friction”, and “saturated friction”, and “sliding” to what’s presented? I could not really find a purpose of “angles”, except to motivate orthogonality — nothing like the arctan appears in the algorithm as far as I can tell?
* It was extremely challenging to understand the actual algorithmic contribution that separated this from BCQ.

Empirically, the results of this paper do not positively distinguish it from TD3 and SAC, two baselines from 2018. Moreover, it is quite concerning this paper does not compare against BCQ, the method on which it is based. The sentence “FQL outperformed baseline methods across multiple tasks with a large margin” is somewhere between misleading and untruthful.

I am pretty unhappy with the quality of this paper from start to finish.

**Questions:**

I had a very hard time understanding the connection between friction and the algorithm, can you possibly clear this up?

---

### Official Review · Reviewer_m6NB · 2025-10-31

**Soundness:** 2
**Presentation:** 3
**Contribution:** 2
**Rating:** 4
**Confidence:** 3

**Summary:**

The paper introduces Frictional Q-learning (FQL), a novel method designed to address the extrapolation error in off-policy
Reinforcement Learning (RL). The central concept draws an analogy to classical mechanics by comparing the extrapolation
error to static friction. In this analogy, unsupported state-action pairs in off-policy RL are described as high-friction
regions, hindering the policy's convergence to the optimal policy of the true Markov Decision Process (MDP).
FQL achieves this by simultaneously pulling the policy towards the replay buffer's data distribution while pushing it away from a
heterogeneous state-action distribution constructed from orthogonal actions. A state-conditioned contrastive VAE (cVAE)
is employed to learn the action distribution within the replay buffer and to generate candidate actions aligned with buffer data.
Empirically, FQL is evaluated on Gymnasium benchmarks, showing competitive performance against standard baselines such
as SAC and TD3.

**Strengths:**

The paper's overall presentation is well-structured, making the core algorithmic ideas relatively easy to follow.
The paper provides a robust theoretical foundation, including derivations and proofs demonstrating that FQL can converge
to the true MDP under specified conditions and that extrapolation error can be controlled. The mathematical
formulations are presented clearly.

**Weaknesses:**

- **Conceptual Justification Deficit of the Static Friction Analogy:** While the analogy to static friction is novel
and provides an intuitive framing, its mathematical grounding and necessity beyond a conceptual metaphor remain tenuous.
The paper heavily leverages this analogy as its main storyline, yet the precise, fundamental mathematical relationship
between the physical concept and the RL problem is not fully established. The authors should more explicitly justify how
this analogy directly informs the algorithm's design choices. Is it merely illustrative, or does it offer unique insights
that guide the formal development of FQL? Without this, the analogy is perceived as more "hand-wavy" than foundational.
- **Ambiguity and Justification of Orthogonal Actions:** The paper utilizes "orthonormal actions" to construct a
heterogeneous state-action distribution, which is central to the "pushing away" aspect of FQL. However, the specific
rationale for choosing Euclidean orthogonality as a measure/proxy for "heterogeneous" or "unsupported" actions requires
more rigorous justification. Are actions orthogonal in Euclidean space inherently "heterogeneous" or "unsupported" in
the context of learned policies and state-action distributions? The paper should provide a clearer explanation of why
this specific definition of orthogonality is appropriate for identifying out-of-distribution regions relevant to
extrapolation error. Furthermore, the paper claims the cVAE with orthogonal actions is beneficial compared to a standard
VAE (as used in BCQ [1]). A detailed explanation is needed on how these orthogonal actions specifically contribute to this
benefit. What precise mechanism do they employ to accelerate convergence or enhance stability that a standard VAE,
potentially with a perturbation model, cannot achieve?
- **Error in Algorithm 1**: There is an error in the formulation of the targets $y_{i+1}$ in Algorithm 1. The reward $r_t$
of the transition is not used.
- **Limited and Potentially Overstated Experimental Results:** The experimental evaluation is conducted on only five
Gymnasium environments, which is a relatively small set for a method claiming broad applicability and state-of-the-art
performance. The statement that "FQL outperformed baseline methods across multiple tasks with a large margin" appears to
be an overstatement given the results presented (e.g., FQL is often "on par" or only marginally better than strong
baselines like SAC and TD3, and sometimes underperforms, as noted in the text for HalfCheetah-v4). To strengthen the
empirical validation, the authors should consider evaluating against more modern off-policy approaches, such as recent
advancements in distributional Q-learning [2] and its implementations like FastTD3 [3], to ensure FQL's standing
against the current state-of-the-art.
- **Incomplete Experimental Setting and Comparison to BCQ:** FQL aims to address extrapolation error in off-policy RL.
However, the experimental setup, where the replay buffer is constantly updated, might not fully stress-test the
algorithm's ability to handle severe extrapolation error, especially in contrast to offline RL settings with fixed
buffers (which BCQ [1] explicitly evaluates). A direct experimental comparison of FQL with BCQ [1], particularly in
settings designed to highlight extrapolation error (e.g., fixed datasets or datasets with significant distributional shift),
is absent. Such a comparison would be crucial for validating FQL's claimed advantages over its direct conceptual predecessor.

[1] Fujimoto et al., 'Off-Policy Deep Reinforcement Learning without Exploration'
[2] Bellemare et al., 'A Distributional Perspective on Reinforcement Learning'
[3] Seo et al., 'FastTD3: Simple, Fast, and Capable Reinforcement Learning for Humanoid Control'

**Questions:**

See my comments under Weaknesses

---

### Official Review · Reviewer_Kgtz · 2025-11-07

**Soundness:** 2
**Presentation:** 1
**Contribution:** 2
**Rating:** 2
**Confidence:** 2

**Summary:**

The authors introduce a new off-policy deep RL algorithm "Frictional Q-learning". This algorithm builds on Batch-Constrained Q-learning by Fujimoto et al. (2019). A major challenge in off-policy RL is the distributional shift between the replay buffer and the policy. The authors take inspiration from classical mechanics to interpret the constraint of staying close to the buffer as a type of static friction.

**Strengths:**

The idea and the method described in the paper are interesting and novel. The authors include a large background section, including a discussion of previous physics-inspired RL, and back up their experimental findings with ablation studies.

**Weaknesses:**

While I am not deeply familiar with the offline and off-policy deep RL literature, I am no novice to deep RL. Nothing in the background section of this paper was particularly new to me. Nevertheless, beginning in section 4 ("Algorithm"), I simply cannot follow the text. It might be that it is written with a different community in mind, maybe people more familiar with offline RL have less trouble here. In this case, the background section should be updated to reflect the necessary prerequisites. For example, the discussion leading to equation (7) focuses on the "exploration error $\mathcal E$". While secion 3.1 describes sources of exploration error, $\mathcal E$ is never defined, and so I cannot parse equation (7). This pattern continues, and so I could not understand what the actual algorithm (FQL) is. If this paper is meant to be understood by the general RL community, the writing needs to be improved considerably. Based on this, I recommend rejection with low confidence.

As I had trouble understanding, I looked up the main prior work, "Off-Policy Deep Reinforcement Learning without Exploration" by Fujimoto, Meger, and Precup (2019). Reading through the background section in that paper, it became obvious that the authors took heavy inspiration from this text. Many parts of the background section, as well as the "Batch Constrained Q-learning" section are slightly reworded copies of sentences from Fujimoto et al. Exact correspondences are equations (4), (5), Theorems 1 to 4, etc. Since this is technically background material, it is of course fine to include prior results. However, the writing is nearly identical in many parts, with slight modifications that to me suggest that the authors wanted to make it more difficult to identify. In parts the copying from Fujimoto et al. even leads to errors, such as the sentence leading into equation (5), which to me made no sense at first (the sentence is about "reweighting", but there is no reweighting  in the equation). However, in Fujimoto et al., the same equation describes "equal weighting", with reweighting discussed in the paragraph following this equation. Based on these observations, I have flagged the submission for an ethics review.

Finally, there are many minor and easily fixable mistakes, most obvious perhaps the inclusion of "conference submissions" in the title, and Figure 1 (III), which shows an unphysical situation (since $f_s \leq \mu_s N$ and $N = 0$). Any situation like this cannot be described as "friction". Here, I also do not understand why Coulomb (1821) is cited to assert that $mg = mg$.

I cannot comment profoundly on the actual contribution of this work, since I could not understand the algorithm or its motivation.

**Questions:**

See weakness section.

**Details Of Ethics Concerns:**

See weaknesses section, paragraph 2. It appears that large parts of the background section of the paper are taken from Fujimoto et al. (2019) and reworded.

---

### Note · Authors · 2025-11-17

I have read and agree with the venue's withdrawal policy on behalf of myself and my co-authors.